# SCAPS Empowered Machine Learning Modelling of Perovskite Solar Cells: Predictive Design of Active Layer and Hole Transport Materials

Mahdi Hasanzadeh Azar [1], Samaneh Aynehband [2], Habib Abdollahi [3], Homayoon Alimohammadi [3], Nooshin Rajabi [3], Shayan Angizi [4], Vahid Kamraninejad [3], Razieh Teimouri [5], Raheleh Mohammadpour [5,*] and Abdolreza Simchi [3,5,*]

1. Department of Engineering Physics, McMaster University, Hamilton, ON L8S 4L8, Canada
2. Department of Physics & Atmospheric Science, Dalhousie University, Halifax, NS B3H 4J5, Canada
3. Department of Material Science and Engineering, Sharif University of Technology, Tehran 14588, Iran
4. Department of Chemistry and Chemical Biology, McMaster University, Hamilton, ON L8S 4M1, Canada
5. Institute for Nanoscience and Nanotechnology, Sharif University of Technology, Tehran 14588, Iran
* Correspondence: mohammadpour@sharif.edu (R.M.); simchi@sharif.edu (A.S.)

**Abstract:** Recently, organic–inorganic perovskites have manifested great capacity to enhance the performance of photovoltaic systems, owing to their impressive optical and electronic properties. In this simulation survey, we employed the Solar Cell Capacitance Simulator (SCAPS-1D) to numerically analyze the effect of different hole transport layers (HTLs) (Spiro, CIS, and $CsSnI_3$) and perovskite active layers (ALs) ($FAPbI_3$, $MAPbI_3$, and $CsPbI_3$) on the solar cells' performance with an assumed configuration of $FTO/SnO_2/AL/HTL/Au$. The influence of layer thickness, doping density, and defect density was studied. Then, we trained a machine learning (ML) model to perform predictions on the performance metrics of the solar cells. According to the SCAPS results, $CsSnI_3$ (as HTL) with a thickness of 220 nm, a defect density of $5 \times 10^{17}$ cm$^{-3}$, and a doping density of $5 \times 10^{19}$ cm$^{-3}$ yielded the highest power conversion efficiency (PCE) of 23.90%. In addition, a 530 nm-$FAPbI_3$ AL with a bandgap energy of 1.51 eV and a defect density of $10^{14}$ cm$^{-3}$ was more favorable than $MAPbI_3$ (1.55 eV) and $CsPbI_3$ (1.73 eV) to attain a PCE of >24%. ML predicted the performance matrices of the investigated solar cells with ~75% accuracy. Therefore, the $FTO/SnO_2/FAPbI_3/CsSnI_3/Au$ structure would be suitable for experimental studies to fabricate high-performance photovoltaic devices.

**Keywords:** perovskite; hole transport layer; solar cell; external quantum efficiency; SCAPS-1D; machine learning



## 1. Introduction

Photovoltaic (PV) cells have achieved significant potential for use in various state-of-the-art applications, including aerospace [1], $CO_2$ reduction [2], green hydrogen production [3], and Agrivoltaic farming [4]. These devices are categorized into three generations based on the materials and employed techniques [5]. The first and second generations mostly include thick silicon crystalline and thin Copper Indium Gallium Di-Selenide (CIGS) and Cadmium Telluride (CdTe) films [6], respectively. The third generation, including dye-sensitized [7], quantum dots [8], and organic solar cells [9–12], is capable of representing a variety of cost-effective, energy-saving thin film systems. However, the main drawback is low efficiency. During the last decade, organic–inorganic perovskites have emerged as one of the advanced PV cells, offering a great opportunity for high PCE with a reasonable processing cost [13–15]. The remarkable performance of perovskite-based solar cells (PSCs) originates from their outstanding optical and electrical properties, such as high absorption efficiency, direct bandgap, long diffusion length, and high charge carrier mobility [16]. Despite the rising trend of PSCs, it is essential to find efficient approaches to maintain the upward trend of

PCE. So far, several practical efforts have been taken to enhance PCE, including controlling the crystallinity and morphology of perovskite films by employing appropriate annealing temperatures and solvents [17,18], tuning the molar composition of the compounds [19], and introducing ionic additives [20]. However, material optimization to further improve PCE has remained a great challenge.

Recently, simulations by SCAPS-1D have enabled researchers to gain a better understanding of the physics of PSCs to optimize different elements of devices precisely toward higher efficiencies [21,22]. Moreover, machine learning (ML) has become a powerful tool for discovering new ways to approach the optimization of solar cells [23]. Thorough reviews have been written about the application of ML in solar cell research [24,25]. It has been demonstrated that ML is useful in predicting the properties and performance of perovskite solar cells.

To increase the performance of PSCs, several factors should be optimized. Among different elements of PSCs, the hole transport layer (HTL) is considered one of the most critical components determining the open-circuit voltage ($V_{OC}$) and efficiency of the cells [26]. Spiro-OMeTAD has been broadly used as HTL, having a vital role in reaching high levels of PCE in over 20% [27]. However, its instability upon exposure to heat and moisture, along with high fabrication costs, has prevented it from being transformed into a commercialized product [28]. Despite many attempts toward replacing organic and inorganic materials, such as PEDOT: PSS [29], $P_3HT$ [30], PTAA [31], $CuSbS_2$ [32], and CuO [33], with Spiro, finding a cost-effective and efficient HTL has remained a great problem.

In addition to HTL, the active layer (AL) is shown to effectively promote PSCs' efficiency. This can be achieved by optimizing the AL characteristics, such as type, doping density ($N_A$), defect density ($N_t$), and thickness. For instance, by taking advantage of SCAPS-1D software, the $N_A$ ($10^{16}$ $cm^{-3}$), $N_t$ ($10^{12}$ $cm^{-3}$), and thickness (700 nm) of a simulated cesium lead iodide ($CsPbI_3$)-based solar cell were optimized, yielding an efficiency of 21.31% [34]. In another survey, the efficiency of a methylammonium lead iodide ($MAPbI_3$)-based solar cell could reach the efficiency of about 21.42% by optimizing the thickness (500 nm) and $N_t$ ($10^{13}$ $cm^{-3}$) of the active layer [35]. Recently, formamidinium lead iodide ($FAPbI_3$) perovskites have received much attention as the most promising candidate in optoelectronics, especially solar cells. This is mainly because, compared to $MAPbI_3$ ($E_g$ = 1.55 eV) and $CsPbI_3$ (1.73 eV), $FAPbI_3$ has the optimum bandgap value near the infrared region ($E_g$ = 1.48–1.51 eV), and affords improved thermal stability along with superior dark storage stability in the device [36,37]. However, only a few experimental investigations have been carried out on $FAPbI_3$-based solar cells, and they have not been comprehensively simulated in the literature.

This study aims to introduce an efficient PSC structure comprised of cost-effective and promising hole transport and active layers. To achieve this, we employed SCAPS-1D software to carry out a comprehensive simulation study. At first, we simulated the performance of PV cells based on different HTLs (CIS and $CsSnI_3$, Spiro-OMeTAD) and ALs ($FAPbI_3$, $MAPbI_3$, and $CsPbI_3$ perovskites). Second, the characteristics of the best candidates, including layer thickness, $N_t$, and $N_A$, were elaborated and optimized. Finally, as one of the recent surveys, we trained an ML model to perform predictions on the performance metrics of the solar cells. This approach can be a robust starting point for further artificial intelligence (AI) driven research and investigations. The novelty of this part lies in combined ML practice and SCAPS-1D supported by experimental research to design predictable high-performance PSCs.

## 2. Materials and Methods

A schematic demonstration of the configured solar cell device used in this survey is shown in Figure 1. As seen, the layers' arrangement from bottom to top is FTO/ETL/perovskite (active layer)/HTL/Contact. The physical parameters were derived from the literature and are summarized in Tables 1 and 2.

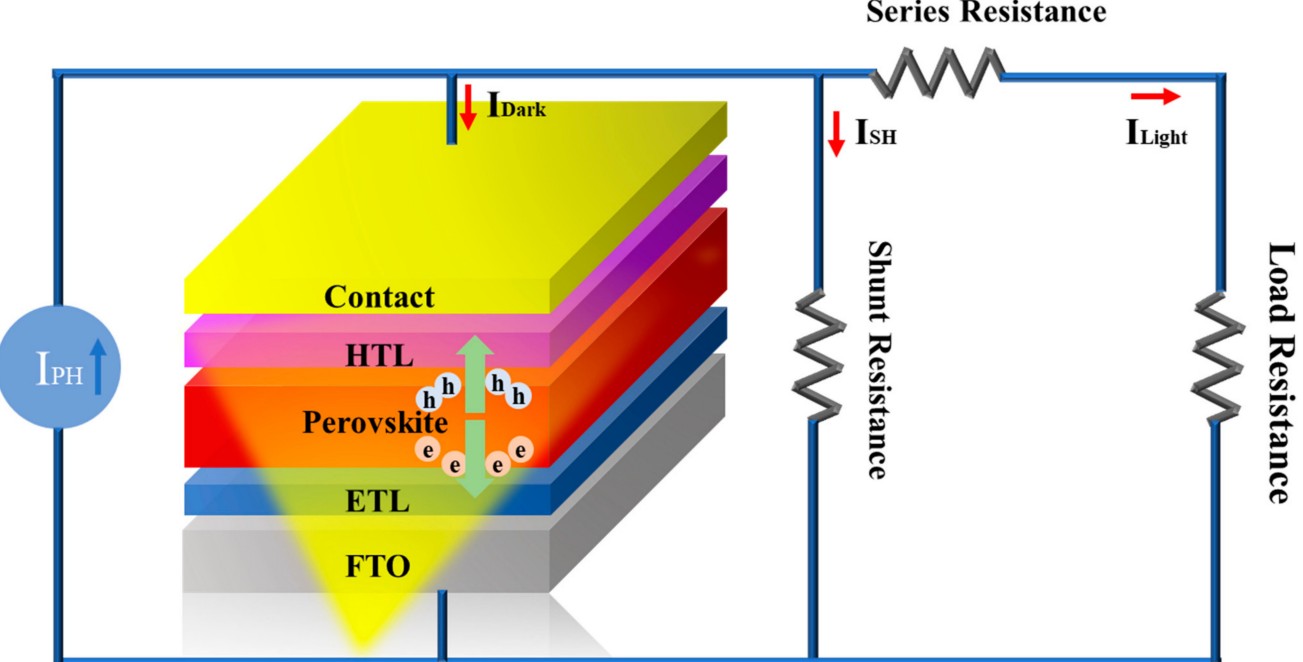

**Figure 1.** Configuration of perovskite solar cells used for simulations.

**Table 1.** Cell parameters used for simulations.

| Parameter | FAPbI$_3$ | MAPbI$_3$ | CsPbI$_3$ |
|---|---|---|---|
| Thickness (nm) | 550 | 550 | 550 |
| E$_g$ (eV) | 1.51 | 1.55 | 1.73 |
| $\chi$ (eV) | 4 | 3.9 | 3.95 |
| $\varepsilon_r$ | 6.6 | 6.6 | 6.6 |
| N$_C$ (1/cm$^3$) | $1.2 \times 10^{19}$ | $1.2 \times 10^{19}$ | $1.2 \times 10^{19}$ |
| N$_V$ (1/cm$^3$) | $2.9 \times 10^{18}$ | $2.9 \times 10^{18}$ | $2.9 \times 10^{18}$ |
| $\mu_n$ (cm$^2$/Vs) | 2.7 | 0.5 | 16 |
| $\mu_p$ (cm$^2$/Vs) | 1.8 | 0.5 | 16 |
| N$_D$ (1/cm$^3$) | $1.3 \times 10^{16}$ | $1.3 \times 10^{16}$ | $1.3 \times 10^{16}$ |
| N$_A$ (1/cm$^3$) | $1.3 \times 10^{16}$ | $1.3 \times 10^{16}$ | $1.3 \times 10^{16}$ |
| N$_t$ (1/cm$^3$) | $1.5 \times 10^{14}$ | $1.5 \times 10^{14}$ | $1.5 \times 10^{14}$ |
| Reference | [38] | [39] | [34] |

**Table 2.** Characteristics of ETL and HTL used for simulations.

| Parameter | SnO$_2$ | Spiro-OMeTAD | CIS | CsSnI$_3$ |
|---|---|---|---|---|
| Thickness (nm) | 90 | 200 | 200 | 200 |
| E$_g$ (eV) | 3.5 | 2.9 | 1.5 | 1.3 |
| $\chi$ (eV) | 4 | 2.2 | 3.55 | 3.95 |
| $\varepsilon_r$ | 9 | 3 | 13.6 | 9.93 |
| N$_C$ (1/cm$^3$) | $2.2 \times 10^{17}$ | $2.2 \times 10^{18}$ | $1 \times 10^{19}$ | $1 \times 10^{19}$ |
| N$_V$ (1/cm$^3$) | $2.2 \times 10^{17}$ | $2.2 \times 10^{18}$ | $1 \times 10^{18}$ | $1 \times 10^{18}$ |
| $\mu_n$ (cm$^2$/Vs) | 20 | $1 \times 10^{-4}$ | 25 | 1500 |
| $\mu_p$ (cm$^2$/Vs) | 10 | $1 \times 10^{-4}$ | 25 | 585 |
| N$_D$ (1/cm$^3$) | $10^{15}$ | 0 | 0 | 0 |
| N$_A$ (1/cm$^3$) | 0 | $1.3 \times 10^{18}$ | $1.3 \times 10^{18}$ | $1.3 \times 10^{18}$ |
| N$_t$ (1/cm$^3$) | $10^{18}$ | $10^{15}$ | $10^{15}$ | $10^{15}$ |
| Reference | [38] | [38] | [32] | [40] |

### 2.1. SCAPS 1-D

SCAPS-1D (version: 3.3.07) is a one-dimensional solar cell simulation software that can numerically solve three differential equations known as continuity and Poisson's equations for electrons and holes. This software can predict device characteristics such as current density–voltage curve, quantum efficiency, energy bands, and other specific responses of the planar thin-film structure under illumination [40]. To examine the performance of configured solar cells, different types of HTLs (Spiro-OMeTAD, CIS, and $CsSnI_3$) and perovskites ($FAPbI_3$, $MAPbI_3$, and $CsPbI_3$) were simulated through a comparative survey. Then, the effects of thickness, $N_t$, and $N_A$ of the active and hole transfer layers on PCE were studied. To improve the real-life application of this simulation, the adverse effect of the charge carrier recombination at interfaces on the efficiency of cells was taken into account. The defect parameters of the thin interface between ETM/perovskite and perovskite/HTL were extracted and summarized in Table 3.

**Table 3.** Defect parameters of ETM/perovskite and perovskite/HTL interfaces, as extracted from Ref. [41].

| Interface | Defect Type | $A_e$ (cm$^2$) | $A_h$ (cm$^2$) | Energetic Distribution | $E_t$ | $E_f$ (eV) |
|---|---|---|---|---|---|---|
| ETM/$FAPbI_3$ | Acceptor | $1 \times 10^{-17}$ | $1 \times 10^{-18}$ | Single | Above the highest EV | 0.32 |
| $FAPbI_3$/HTL | Acceptor | $1 \times 10^{-18}$ | $1 \times 10^{-19}$ | Single | Above the highest EV | 0.07 |

$A_e$: Capture cross-section electrons; $A_h$: Capture cross-section holes; $E_t$: Reference for defect energy level; $E_f$: Energy level concerning the reference.

### 2.2. Machine Learning

Machine learning was performed using Python, a high-level programming language famous for its diverse use in AI [42–45]. Numpy and Pandas were used to work with the dataset. The dataset consists of 60 rows and 33 columns, each representing a feature of a solar cell. The data were cleaned, and the values of each column were converted to an ML learning-compatible format. The Pearson correlation coefficient was used to remove highly correlated features. This approach is common because the features are supposed to be independent of each other, and a high Pearson correlation coefficient violates this assumption. The employed equation for this coefficient is [46]:

$$r = \frac{\sum(x_i - \bar{x})(y_i - \bar{y})}{\sqrt{(x_i - \bar{x})^2 \sum(y_i - \bar{y})^2}} \tag{1}$$

Scikit-Learn was used to build a simple Random Forest at first to decide the future presence of each feature in the final model. This model helped to exclude features with the least importance and impact. Eleven features were selected after this comprehensive phase. Pymatgen and Matminer packages were used for feature extraction on the composition of perovskites. The data were split into train and test categories for model training. The final model, which was another random forest, was initialized with optimized hyperparameters. Two separate ways were considered to determine the best hyperparameters for the Random Forest model. First, a parameter grid (a dictionary in Python) containing a list of different values for each parameter as the key was prepared. From the Scikit-Learn tools, the RandomSearchCV and GridSearchCV were used to evaluate the best combination of the aforementioned grid as the selected hyperparameters. Random search performs by randomly selecting a combination from the available options in the grid and evaluating the model's performance via CV. On the other hand, Grid Search evaluates every possible combination of the parameters provided in the grid (which logically should provide better results than the Random Search would). Therefore, GridSearchCV was used to evaluate the best combination of hyperparameters from a range of different values for each parameter. The final model was evaluated with five-fold cross-validation to ensure

its performance and consistency. Finally, a test set was employed to evaluate the model's accuracy and performance.

By aggregating the result of each decision tree, the random forest reduces the chance of overfitting and improves the overall predictive performance [46]. Overfitting is the state where the model tends to reduce its bias to the training data, which causes increased variance on the test (and/or validation) set, indicating the poor predictive performance of the model. This issue can be addressed by comparing the prediction results of the train set to the test set, which can be handled in various ways, including increasing the number of decision trees, decreasing each one's maximum depth, and using a subset of features on each split, and so forth. It is worth mentioning that the evaluation metric was the coefficient of determination of the prediction. According to the Scikit-Learn official website, this metric presented as is defined as:

$$R^2 = (1 - \frac{u}{v}) \tag{2}$$

$u$ is the residual sum of squares $((y_{pred} - y_{true})^2)$. $v$ is the total sum of squares defined as $(y_{pred} - mean(y_{true}))^2$. A constant model that always predicts the expected value of y, disregarding the input features, would achieve an $R^2$ score of 0.0.

The importance of a node j in a single decision tree was calculated by [46]:

$$ni_j = w_j C_j - w_{left(j)} C_{left(j)} - w_{right(j)} C_{right(j)} \tag{3}$$

where $w_j$ is the weighted number of samples in node j as a fraction of the total weighted number of samples. $C_j$ is the impurity in node j and left (j) and right (j) are its respective child nodes. The feature importance of the feature i is also calculated by [46]:

$$fi_i = \frac{\sum_{j:nodejsplitsonfeaturei} ni_j}{\sum_{j \in allnodes} ni_j} \tag{4}$$

Finally, the feature values were averaged over all individual decision trees.

## 3. Results and Discussion

### 3.1. Type of Hole Transport Layer

Among various layers included in a solar cell device, HTL plays an essential role in solar cell performance via two main pathways [47]: (I) it facilitates the transport of photogenerated holes to the counter electrode; (II) HTL prevents the direct interaction between the active layer and the electrode. As mentioned before, despite the extensive integration of Spiro-OMeTAD into solar cells to enhance PCE, the low cost-effectiveness, low hole mobility, and server instability under certain conditions have limited their applications. Therefore, different types of HTLs have been examined. To begin the simulation, a solar cell model of FTO/SnO$_2$ (90 nm)/FAPbI$_3$ (550 nm)/Spiro-OMeTAD (200 nm)/Au was simulated based on an experimental setup to ensure the accuracy of the results [48]. SnO$_2$ was chosen as ETL due to its high stability, mobility, transmittance, energy alignment with perovskite, and facile processing [49]. As is shown in Figure 2, the obtained curve and parameters from simulations are well-fitted with the experimental results, attesting to the reliability of our numerical simulations.

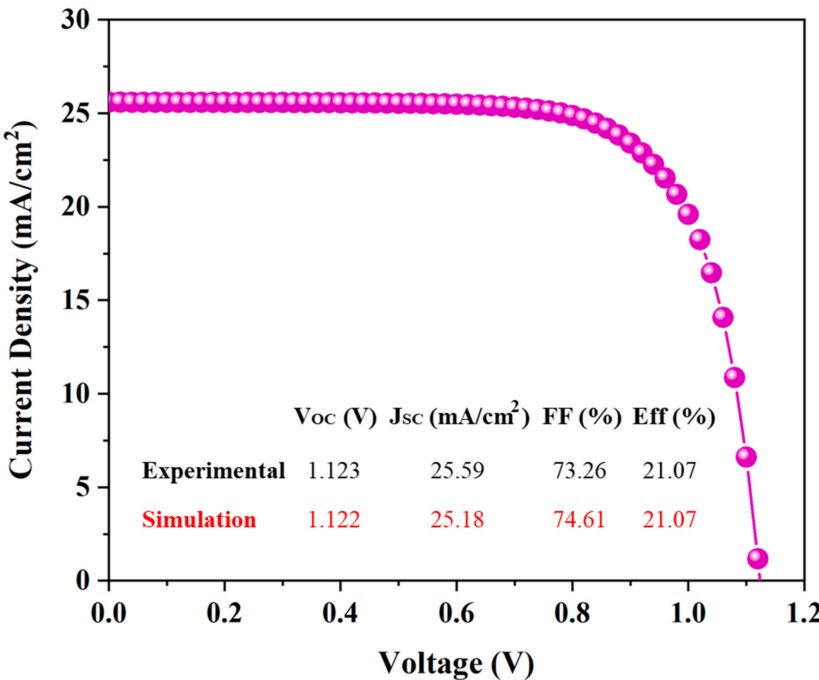

**Figure 2.** J-V curve for a planar perovskite solar cell with a structure of FTO/SnO$_2$ (90 nm)/FAPbI$_3$ (550 nm)/Spiro-OMeTAD (200 nm)/Au. For simulations, the physical parameters are extracted from Tables 1 and 2. The experimental data were adopted from Ref. [48].

To explore the effect of HTL type on the solar cell's performance, different HTLs, including Spiro-OMeTAD, CIS, and CsSnI$_3$, were examined. The thickness of HTL was considered constant (200 nm) for all materials. The results of the simulations are presented in Figure 3a. The figures of merit, including open circuit voltage (V$_{OC}$), short circuit current density (J$_{SC}$), filling factor (FF), and efficiency, are listed in Table 4. The results demonstrate that the J$_{SC}$ of the perovskite solar cell with CIS could reach 26.50 mA/cm$^2$, which is approximately 1 mA/cm$^2$ higher than that of Spiro-OMeTAD. This probably corresponds to not only the broader absorption coefficient but also, the higher carrier velocity of CIS, which accelerates carrier transport, leading to higher carrier density [27]. However, the V$_{OC}$ of the CIS-based cell is lower than that of Spiro. This may be attributed to the larger offset or higher mismatch between the valence band (VB) of the perovskite and the highest occupied molecular orbital (HOMO) of the CIS, which decreases the V$_{OC}$ in PSCs due to hole transport losses [50] (Figure 3b). Overall, the simulated solar cell based on CIS was capable of indicating better efficiency than that of Spiro.

**Table 4.** The effect of HTL on the solar cell parameters.

| HTL | V$_{OC}$ (V) | J$_{SC}$ (mA/cm$^2$) | FF (%) | Efficiency (%) |
|---|---|---|---|---|
| Spiro-OMeTAD | 1.123 | 25.59 | 73.26 | 21.07 |
| CIS | 1.1051 | 26.50 | 73.94 | 21.65 |
| CsSnI$_3$ | 0.935 | 29.5 | 79.86 | 22.02 |

The use of CsSnI$_3$ as an HTL results in attaining a V$_{OC}$ of 0.935 V, J$_{SC}$ of 29.5 mA/cm$^2$, FF of 79.86%, and an overall efficiency of 22.02%. In addition to possessing better figures in the simulation assessment, in the real experiments, CsSnI$_3$ is expected to be a better candidate as it has a similar unit cell, structure, and lattice constant with the active layer, which can cover dangling bonds and interface defects to decline the recombination sites [51,52]. However, its lower V$_{OC}$ is a challenge that will be optimized in this paper by tuning doping and defect density. First of all, the lower V$_{OC}$ of CsSnI$_3$ compared to other layers can be explained as follows. Regarding the energy difference viewpoint, the energy difference

between the HOMO of $CsSnI_3$ (−5.25 eV) and perovskite (−5.5 eV) is lower than that of other layers (or there is a higher difference between the quasi-Fermi level of $CsSnI_3$ and ETL) which should have shown higher $V_{OC}$. This may stem from the low bandgap energy of $CsSnI_3$, as widening the bandgap energy succeeded in increasing the $V_{OC}$.

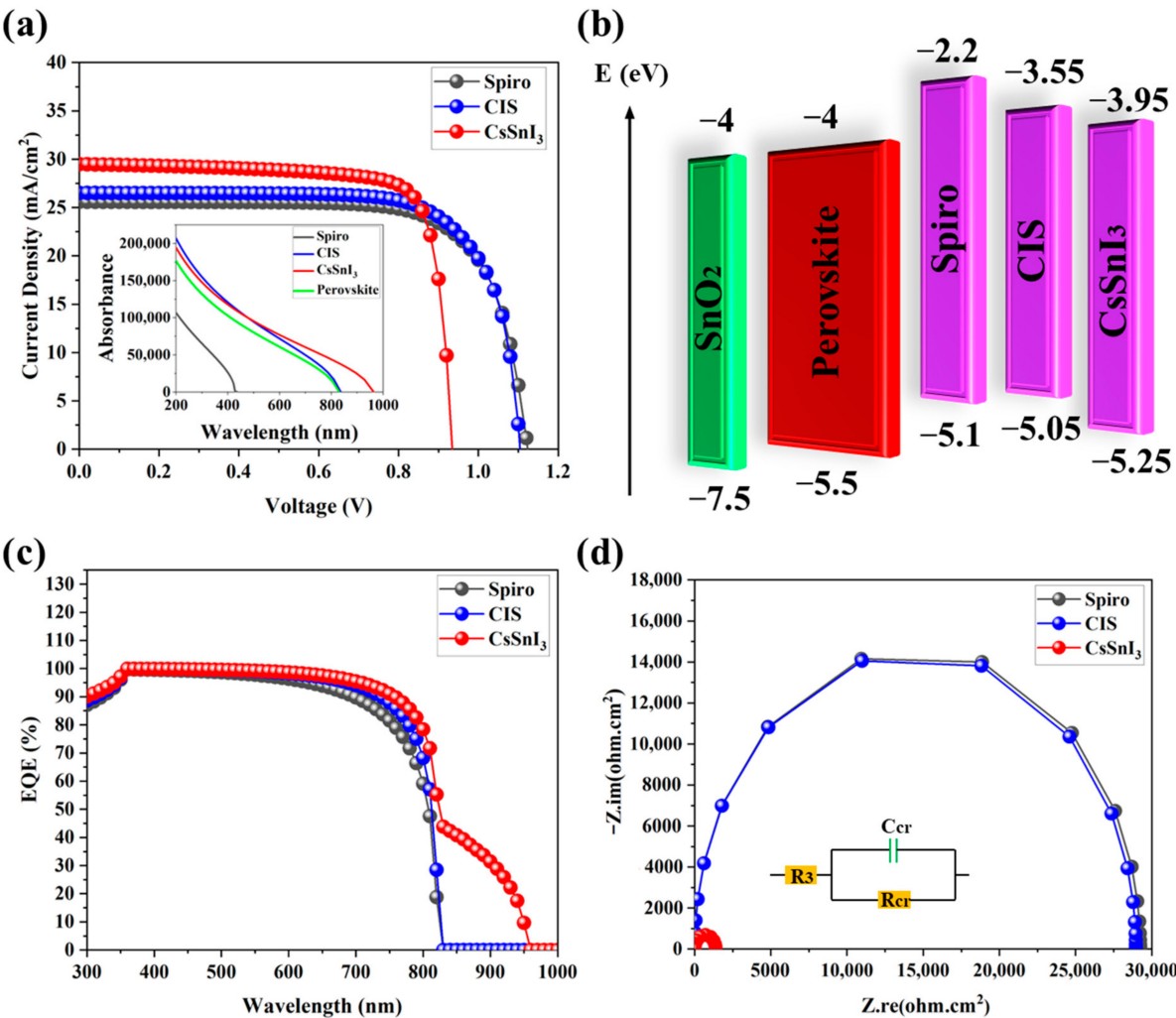

**Figure 3.** (**a**) J-V curves of simulated solar cells with different HTLs (The absorption curves of different HTL layers and the perovskite are shown in the inset). (**b**) The schematic of the energy level diagram of $FAPbI_3$ and the examined HTLs. (**c**) EQE and (**d**) Nyquist plots of the simulated PSCs with different HTLs.

However, the current density, which is affected by the photo-generated minority carrier current [53,54], is considerably higher than the other cells. To elucidate $J_{SC}$, the external quantum efficiency (EQE) of the cells was simulated. Figure 3c determines that EQE is nearly constant in the region from 360 nm to 600 nm, and then it takes a downward trend to 800 nm. It is important to point out that the EQE spectrum of $FAPbI_3$/Spiro and $FAPbI_3$/CIS cells does not exhibit absorption beyond 821 nm (the bandgap energy of the perovskite). Nevertheless, the EQE of $FAPbI_3$/$CsSnI_3$ shows the capability of absorbing a wider range of the spectra, particularly in the near-infrared region (see the inset in Figure 3a). $CsSnI_3$ has a lower bandgap energy (1.3 eV) than $FAPbI_3$ (1.51 eV) [55], thus contributing to the absorption. Moreover, the higher hole mobility decreases the probability of recombination, resulting in a noticeably higher short-circuit current and FF [56]. This better charge transfer and capacity interface can be found by electrochemical impedance spectroscopy [57]. Nyquist plots in Figure 3d for perovskite solar cells containing Spiro-

OMeTAD, CIS, and $CsSnI_3$, as $CsSnI_3$ has the smallest arc, which indicates better transfer conductivity. Overall, the simulation results determine the potential of $CsSnI_3$ as an HTL to attain efficient perovskite solar cells empowered by a more effective absorption capacity and significant hole mobility.

### 3.2. Effect of $CsSnI_3$

3.2.1. Thickness

In the previous section, it was shown that $CsSnI_3$ is a promising material as an HTL. As the characteristics of an HTL, including thickness, doping density, and defect density, significantly influence the performance of solar cells, these parameters must be optimized. For instance, HTL thickness is essential in effective carrier transport and controlling the recombination rate [58]. Therefore, to attain efficient energy conversion, the thickness of an HTL should be optimized.

Figure 4a,b show the effect of $CsSnI_3$ thickness on the $J_{SC}$ and $V_{OC}$ of the perovskite solar cells. The results indicate that $J_{SC}$ and $V_{OC}$ are improved with an increased $CsSnI_3$ thickness up to 220 nm. According to experimental surveys [59], thickening the HTL gives rise to smoother film surfaces with reduced interfacial recombination that ultimately improves $V_{OC}$. In addition, a thicker HTL reduces the roughness of the Au layer with enhanced light reflectivity, boosting carrier generation and, thus, the current density [59,60]. Since, in our simulations practice, roughness is considered independent and constant, we sought other reasons for this observation. First, the capacity of a thinner HTL is not sufficient to accumulate whole holes. The small number of accumulated holes also pushes a backward force to incoming holes and decreases the effective carriers. Second, the depletion region between the perovskite layer and a thin HTL is not as powerful as thicker ones, which fails to provide a strong driving force for the separation of carriers. The results have also indicated that when the thickness of HTL is higher than 220 nm, $V_{OC}$ and $J_{SC}$ are prone to diminish. This is mainly because the traveling distance of the carriers to arrive at the interface increases. As a result, traps encourage non-radiative Shockley–Read–Hall recombination and the augmented trap-assisted recombination rate [61]. The effect of HTL thickness on FF is shown in Figure 4c. The first downward trend is a consequence of an increase in series resistance and charge career transport time. By reaching the HTL thickness of around 260 nm, the layer fully covers the perovskite film, raising the shunt resistance with improved FF [59,60]. The variation in PCE with the HTL thickness indicates an optimum $CsSnI_3$ thickness of 220 nm, which yields PCE = 22.12% (Figure 4d).

3.2.2. Doping Density

Studies have determined that the doping density ($N_A$) of the HTL greatly influences the characteristics of the PV units, including conductivity, recombination rate, $V_{OC}$, and the diffusion length of the carriers [56]. Therefore, the parameters of solar cells at various doping densities in the range of $10^{14}$ to $10^{21}$ $cm^{-3}$ at a constant HTL thickness (200 nm) were simulated [62]. Figure 5a shows that $V_{OC}$ increases by increasing $N_A$. It is obvious that the Fermi level, which is the electrochemical potential of the electron in a solid, is typically placed near the conduction band for n-type semiconductors and the valence band for p-type semiconductors. In layered semiconductor structures, the Fermi level is lined up at the same value and reaches an equilibrium state, causing band bending and built-in voltage ($V_{bi}$) [63]. Because of this internal electric field, the dissociated photogenerated electron and holes drift towards the n and p regions, respectively, and accumulate there. Under illumination and open circuit conditions, the excited carriers cause splitting electron and holes quasi Fermi-levels ($E_{Fn}$ and $E_{Fp}$, respectively) in the photoactive semiconductor material, canceling $V_{bi}$, and forming a photo-voltage called $V_{OC}$, which can be written as follows [64]:

$$V_{OC} = \frac{1}{q}(E_{Fn} - E_{Fp}) \tag{5}$$

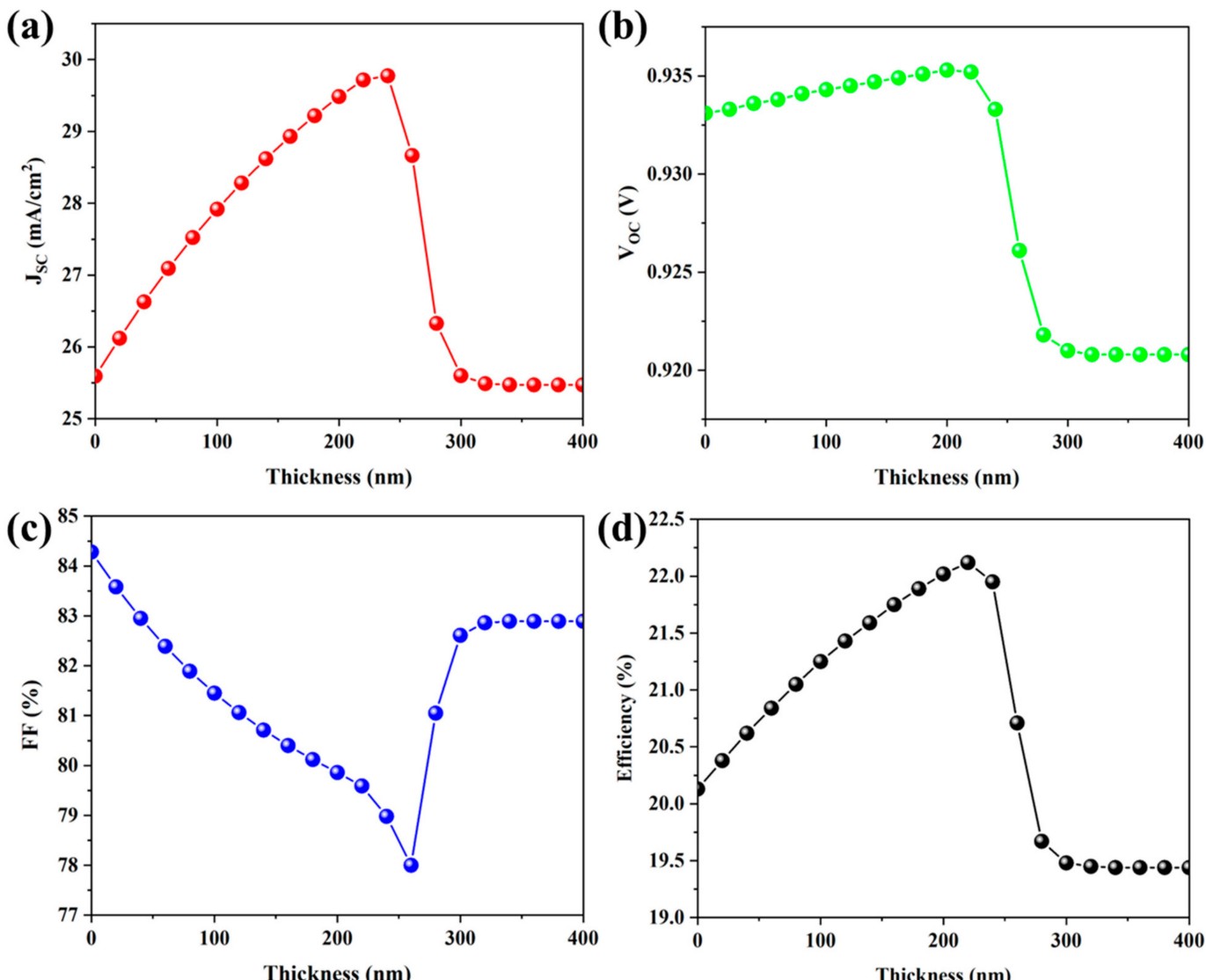

**Figure 4.** Effect of CsSnI$_3$ thickness on the performance of the studied perovskite solar cells. (**a**) J$_{SC}$, (**b**) V$_{OC}$, (**c**) FF, and (**d**) PCE.

The energy band diagrams at doping concentrations of $5 \times 10^{14}$ and $5 \times 10^{20}$ cm$^{-3}$ are shown in Figure 5b,c, respectively. The doping of HTL shifts down the Fermi level and increases the difference between the quasi-Fermi levels of CsSnI$_3$ and SnO$_2$ through the perovskite to build a higher V$_{OC}$. Moreover, increasing the dopant concentration reduces the recombination rate as a result of enhanced internal electric field and accelerated carrier separation. Meanwhile, J$_{SC}$ decreases slightly at the N$_A$ level ranging from $10^{14}$ to $10^{18}$ cm$^{-3}$. A significant decline in the current density is achieved at $10^{20}$ cm$^{-3}$ because numerous deep Coulomb traps have been generated that decrease the hole mobility [65,66]. Figure 5d shows that the highest PCE (23.38%) is obtained when the dopant concentration reaches $\sim 5 \times 10^{19}$ cm$^{-3}$.

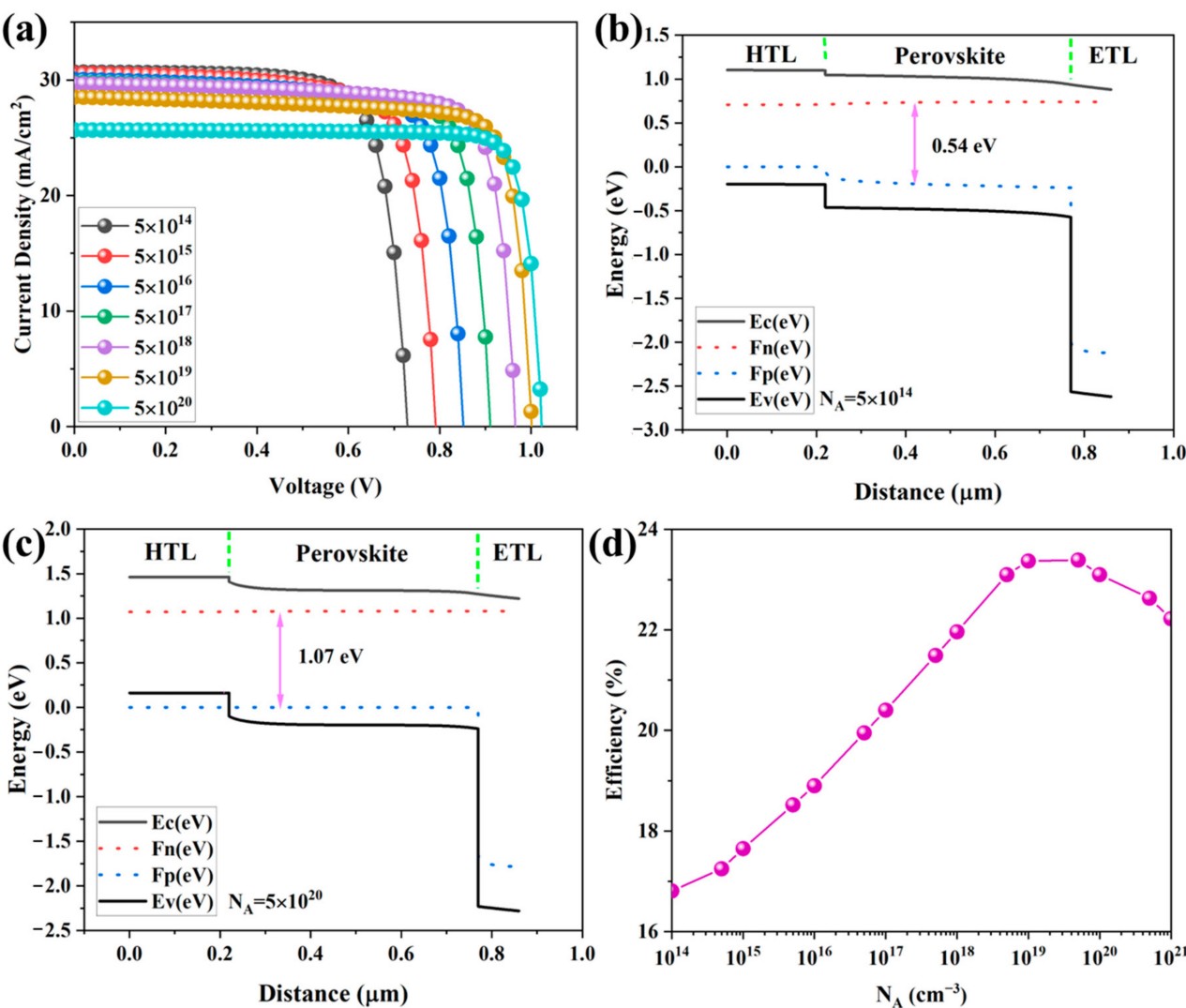

**Figure 5.** (**a**) J-V curves of perovskite solar cells depending on the doping density of the CsSnI$_3$ layer. (**b**,**c**) Band diagram of the cells at the doping densities of $5 \times 10^{14}$ and $5 \times 10^{20}$ cm$^{-3}$. (**d**) Effect of doping density on PCE.

### 3.2.3. Defect Density

In addition to the doping density, HTL defect density (N$_t$) influences the performance of perovskite solar cells. Figure 6 shows this effect on the characteristics of the investigated cells. In the range of $10^{15}$ to $10^{17}$ cm$^{-1}$, no major changes in the characteristic parameters, including V$_{OC}$, J$_{SC}$, and PCE, are noticed (Figure 6a,b,d). However, a sharp decline in VOC and JSC is seen at higher Nt values. This change is mainly attributed to the instant generation of abundant recombination sites in the HTL and interfaces [67]. A higher HTL defect density induced by different sources, such as unwanted foreign atoms, native defects, and dislocations, creates shallow or deep traps. Therefore, as non-radiative recombination centers, the traps are harmful to cell performance. The inset of Figure 6d demonstrates that with increasing the number of interface defects, a lower efficiency is attained. These defects, which mostly stem from the lattice mismatch between AL and HTL and/or iodide or MA migration, form deep traps at the interface and act as Shockley–Read–Hall (SRH) recombination centers. However, in the case of CsSnI$_3$ and FAPbI$_3$, the similarity of the lattice constants declines the probability of interface defects generation [40,61]. As a result, PCE declined to about 21.5% at N$_t$ = $10^{21}$ cm$^{-3}$. Similarly, the initial downward trend of FF is owed to the high number of recombination centers and the amount of series resistance caused by the significant number of traps (Figure 6c). When N$_t$ exceeds $10^{18}$, an

upward trend is observed. This behavior can be attributed to the tunneling effect, which happens when $N_t$ substantially rises, i.e., traps are as close together as to prefer to adopt the tunneling method of transport [68]. Notwithstanding the better results in lower defect density, the simulation results deviate from the experiments; hence, $5 \times 10^{17}$ was selected.

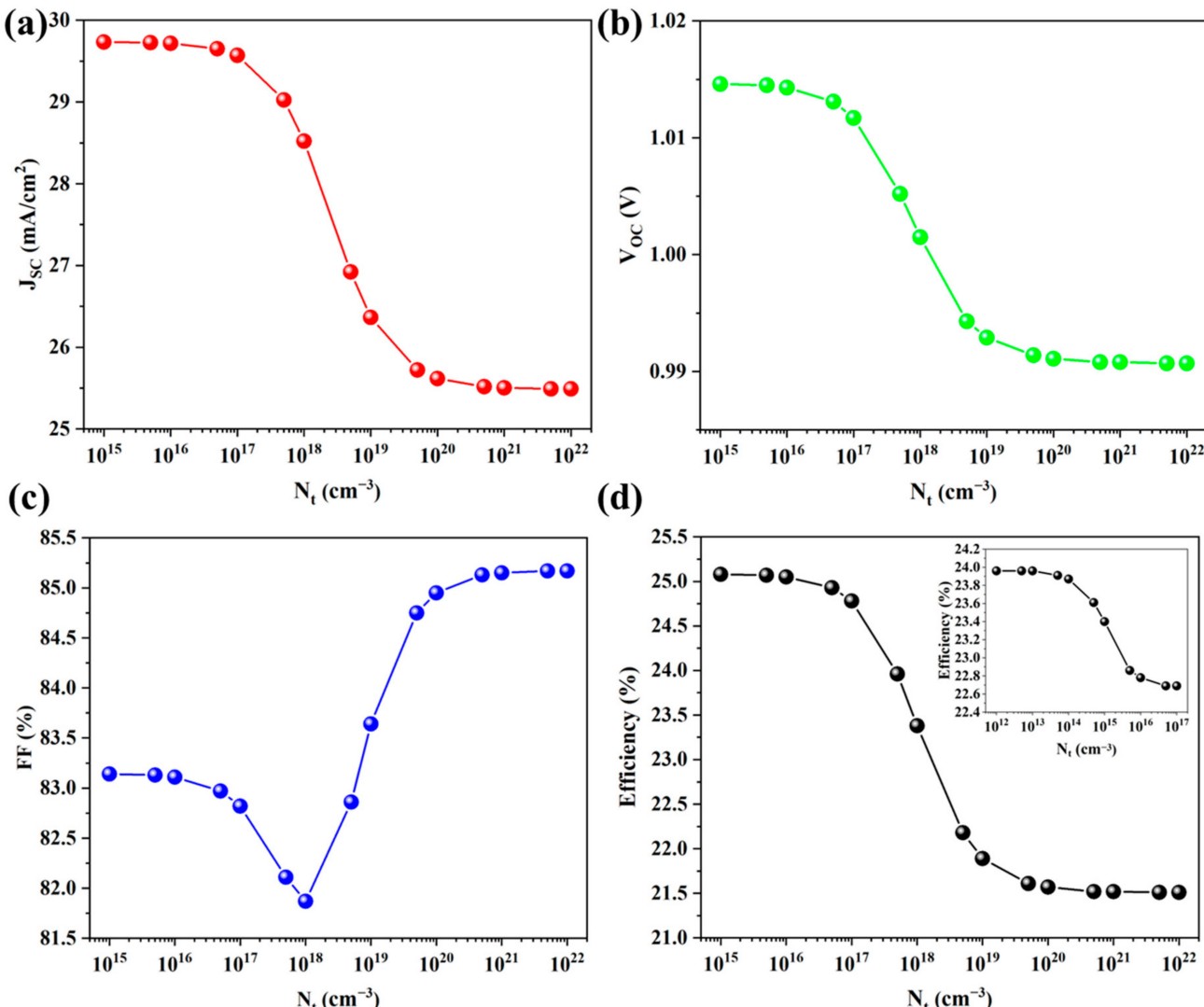

**Figure 6.** Effect of defect density of HTL on (**a**) $J_{SC}$, (**b**) $V_{OC}$, (**c**) FF, and (**d**) PCE of perovskite solar cell with $CsSnI_3$ (The inset shows the effect of interface defect density on PCE).

### 3.3. Active Layer

A solar cell's light absorption and photoelectric conversion mechanism are intertwined with the type of active layer (AL). We investigated this effect for both types of perovskite materials, including inorganic ($CsPbI_3$) and organic–inorganic ($FAPbI_3$ and $MAPbI_3$) compounds. The thickness of AL was 550 nm for all cells. HTL was $CsSnI_3$ with a 220 nm thickness and an $N_A$ and $N_t$ of $5 \times 10^{19}$ and $5 \times 10^{17}$, respectively. The predicated J-V curves and the figures of merit for the simulated cells are shown in Figure 7a and Table 5. The recorded $V_{OC}$ values of $CsPbI_3$, $MAPbI_3$, and $FAPbI_3$ are 1.015, 1.102, and 1.01 V, respectively.

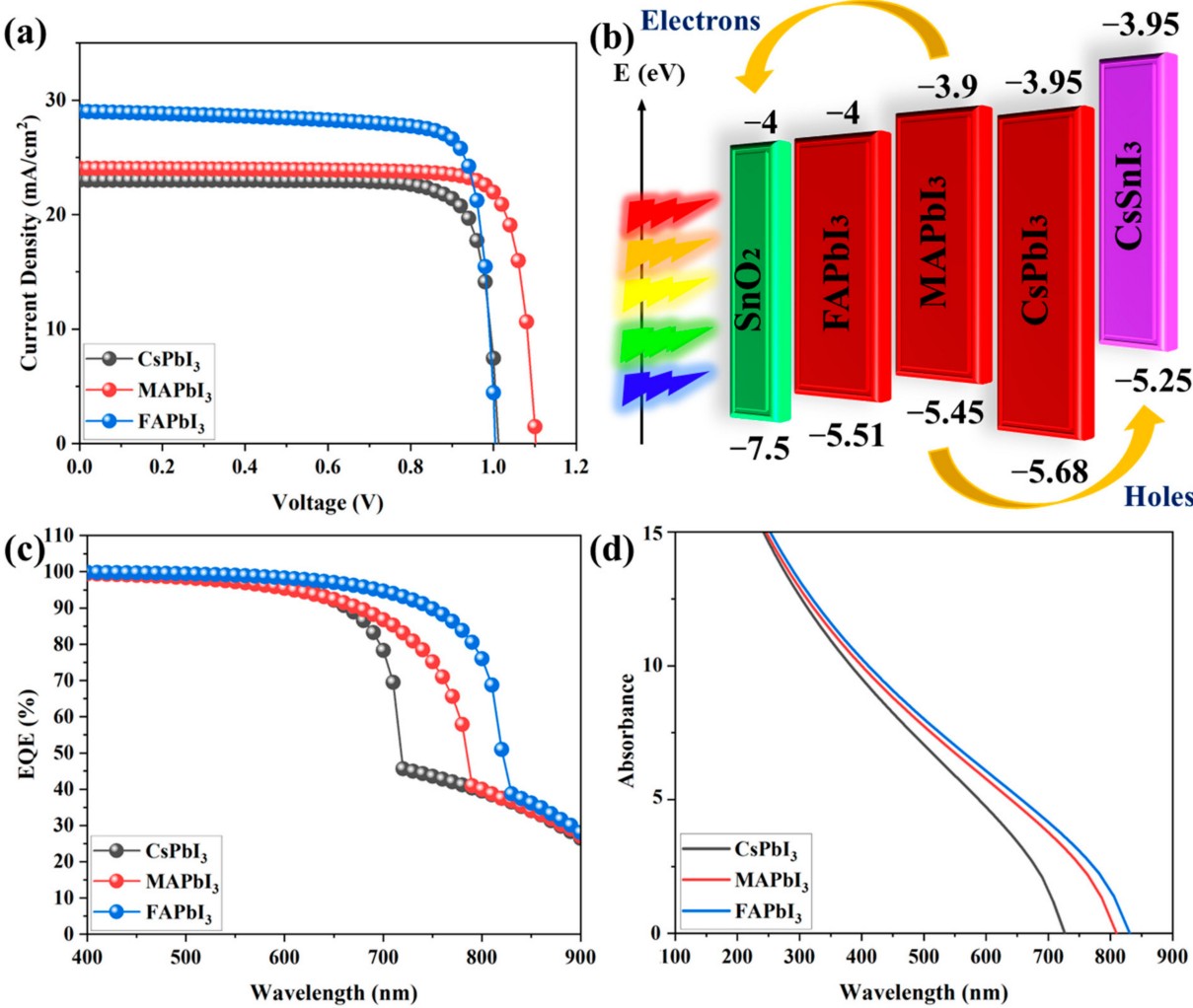

**Figure 7.** (**a**) J-V curves of the planar solar cells made of different perovskites as AL. (**b**) Energy level diagram of FAPbI$_3$, MAPbI$_3$, and CsPbI$_3$. (**c**) EQE and (**d**) absorption diagrams of solar cells with different types of perovskite layers, respectively.

**Table 5.** Effects of perovskite materials on the figures of merit of studied solar cells.

| Perovskite | V$_{OC}$ (V) | J$_{SC}$ (mA/cm$^2$) | FF (%) | Efficiency (%) |
|---|---|---|---|---|
| FAPbI$_3$ | 1.010 | 29.02 | 82.11 | 23.94 |
| MAPbI$_3$ | 1.102 | 24.05 | 83.54 | 22.15 |
| CsPbI$_3$ | 1.015 | 23.02 | 75.33 | 17.56 |

To discuss the obtained results, only the side of the HTL and perovskite is noticed, as the energy difference between perovskite and ETL is almost the same. The highest V$_{OC}$ is related to MAPbI$_3$ because this material has the lowest mismatch with CsSnI$_3$ (Figure 7b) with a bandgap energy value between CsPbI$_3$ and FAPbI$_3$. In terms of CsPbI$_3$, we expect that the higher bandgap energy yields a higher V$_{OC}$, which is not seen for CsPbI$_3$. Despite the advantage of E$_g$ for CsPbI$_3$, the significant energy difference between perovskite and CsPbI$_3$ (4.3 eV) increases mismatch, which contributes to lower V$_{OC}$. In the case of FAPbI$_3$, the lower mismatch between perovskite and CsSnI$_3$ offsets the lower bandgap energy, and the same value of V$_{OC}$ was achieved.

Moreover, the results also determine that the current density of the FAPbI$_3$ solar cell is higher than that of the other ones. Simulations of the quantum efficiency (Figure 7c) reveal that EQE changes at wavelengths beyond 600 nm and is suddenly dropped in the

near-infrared region, i.e., 700 nm (for the inorganic perovskite) and 800 nm (for the organic–inorganic perovskites). However, as $FAPbI_3$ has the most favorable or red-shifted bandgap energy, it is capable of absorbing photons until the wavelength of 830 nm that other active layers are unable (Figure 7d). In addition, $FAPbI_3$ has higher carrier velocity than $MAPbI_3$, which increases the carrier lifetime and decreases the non-radiate recombination. Of particular interest, $FAPbI_3$ exhibits a high EQE in the near-infrared region with a maximum PCE of 23.94% (Table 5). In the aspect of stability of solar cells, the alpha phase of $FAPbI_3$ perovskite with impressive optical and electrical properties is more moisture-resistant and current-stable than $MAPbI_3$, which is mainly originated from the fact that it has more hydrogen bonds and a larger radius cation than $MAPbI_3$ perovskite, making it more stable against destructive objectives and ion migration [36]. As I and MA vacancies (the positively and negatively charged ions) migrate and accumulate at interfaces, the built-in electric field or extraction driving force is weakened due to the generation of an intensive internal electric field, contributing to normal hysteresis [69].

### 3.3.1. Effect of Thickness

The performance and efficiency of perovskite solar cells greatly depend on the thickness of the active layer. We studied the effect of AL thickness on the performance of $FTO/SnO_2$ (90 nm)/$FAPbI_3$/$CuSnI_3$ (220 nm)/Au. Figure 8a determines that PCE gradually increases with increasing the AL thickness up to 530 nm because as more charge carriers are generated by the absorbed photons, higher EQE and current density and, thus, a better performance is attained [59,60]. The fingerprint of this enhancement can also be seen in the J-V curves (Figure 8b). Further thickening of AL slightly diminishes $J_{SC}$ due to the augmented trap-assisted recombination rate. Although there is no considerable change in $V_{OC}$ by increasing the thickness until 590 nm, it starts to drop at higher thicknesses. Increasing the traveling distance of carriers towards the interface of HTL and ETL contributes to more non-radiative Shockley–Read–Hall recombination. In other words, two phenomena, including the higher density of charge carriers and enhanced recombination rate, compete for affecting PCE depending on the AL thickness [70]. We have found that an AL thickness of 530 nm yields the highest efficiency (24.04%).

### 3.3.2. Effect of Defect Density

The J-V response of PSCs depending on the defect density ($N_t$) of AL is simulated and presented in Figure 8c. Similar to the HTL effect, no significant change in the curves up to $N_t = 10^{13}$ cm$^{-3}$ is noticed. However, the figures of merit, including $J_{SC}$, $V_{OC}$, FF, and PCE, decrease at a high concentration of defects. Similar to the defect of HTL, the higher concentration of defects in the low-quality perovskite layer would increase the number of non-radiative recombination centers to degrade the cell performance, which is attributed to changes in the charge carriers' diffusion length and carriers' lifetime. Although a low AL defect density may contribute to higher efficiency in the simulated solar cells, the high fabrication cost is distracting. A low defect density does not also resemble real-life conditions because these types of perovskites are unstable under humid conditions and partly degraded. Therefore, we propose that $N_t = 10^{14}$–$10^{15}$ cm$^{-3}$ is used to manufacture high-performance solar cells with reasonable stability. The best-simulated cell exhibits a PCE of 24.1% with $J_{SC} = 29.00$ mA/cm$^2$, $V_{OC} = 1.01$, and FF = 82.50 (Figure 8d). Compared with the experimental study of Lin et al. [34] on $FTO/SnO_2/CsPbI_3/CuO_2/Au$ solar cells, it appears that tuning the thickness and doping density of $FAPbI_3$ and utilizing $CsSnI_3$ as HTL significantly improves PCE by about 2.5%.

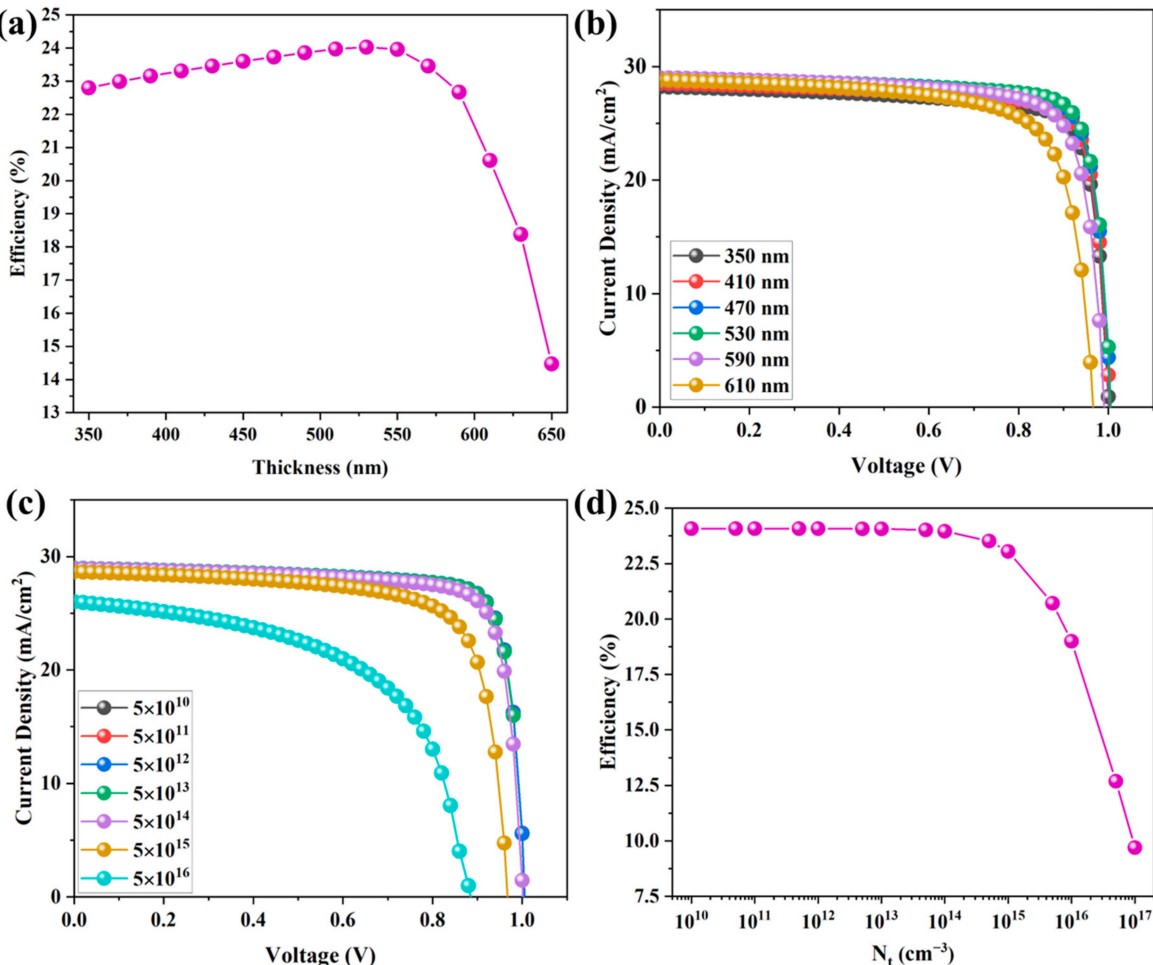

**Figure 8.** (**a**) Efficiency of solar cells versus thickness of the active layer. (**b**) J-V characteristics of solar cells depending on the FAPbI$_3$ thicknesses. Effects of defect density on the (**c**) J-V response and (**d**) efficiency of the perovskite solar cells.

### 3.4. Machine Learning

In the present work, the Random Forest algorithm was employed for the training and testing phases because of its robustness against overfitting and optimum complexity [45]. It is worth noting that more powerful contenders, such as neural networks and deep neural networks, introduce unnecessary complexity. We believe that the added complexity will cause overfitting and eventually lead to poor performance [71,72]. "Kolmogorov Complexity" is a concept that denotes when the complexity and cardinality of the data are greater than the model's ability to comprehend, even large amounts of data may not increase the model's accuracy. In our case, the size of the dataset was too small, and the complexity of the data definitely demanded more records to be added. Previous studies have indicated that reducing the complexity of the dataset can potentially improve the accuracy of the model [73,74]. Considering the low number and high complexity and cardinality of the data points, focusing on reducing this complexity can lead to far better results. By employing Scikit learn, we could observe the effect of every single feature in the overall predictive performance of the ML model. The relative importance of the 11 features with the highest importance is shown in Figure 9a. The common features between perovskite and HTL layers are indexed in the respective order; index 2 denotes an attribute of the HTL layer. The results reveal that thickness, N$_t$, and N$_A$ of HTL are the most responsible features that dictate the difference in the performance of two distinct solar cells. This valuable information can optimize the workflow and individual steps toward designing new solar cells. Therefore, these features were selected to build a new model

with approximately identical performance but a noticeably less size due to the difference in feature number. Model predictions were calculated and plotted for each target value, as shown in Figure 9b–e. The plots must have a slope as close to 1 as possible. The prediction of the ML model determines a mean accuracy of 75% with a mean $R^2$ of 0.78 between actual and predicted values per target feature. Although the accuracy is not high, this achievement would serve as a commendable gateway toward further investigations and a much-needed bridge between the fields of artificial intelligence and materials science. Undoubtedly, a fully trained model constructed on a wide range of input data (trained values) would result in more accurate and realistic outcomes.

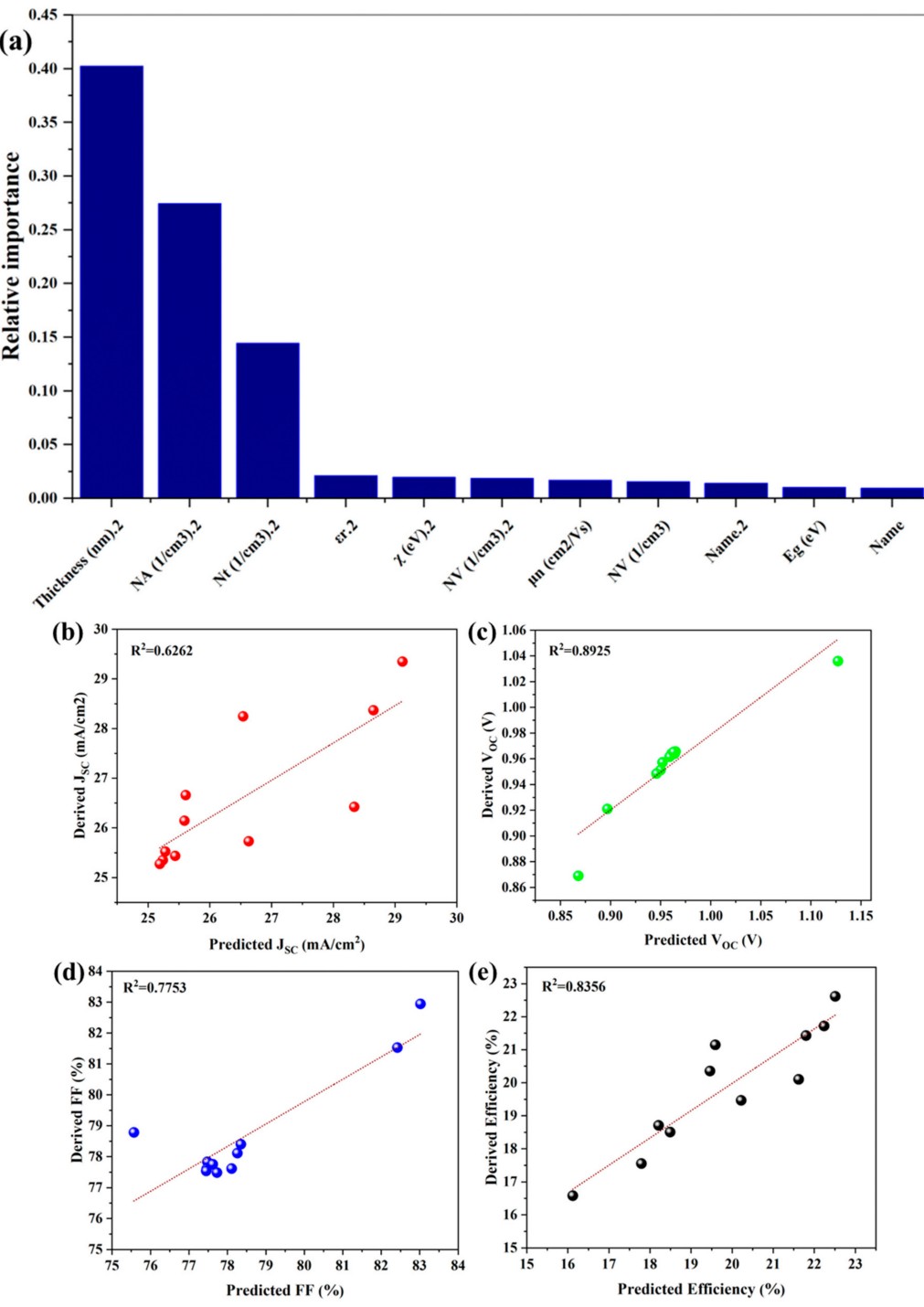

**Figure 9.** (**a**) Various features with the highest importance in the ML model. Derived and predicted values of (**b**) $J_{SC}$, (**c**) $V_{OC}$ (**d**) FF (**e**) PCE.

## 4. Conclusions

In summary, we used SCAPS-1D to evaluate the effect of different parameters on the performance of perovskite-based solar cells through comprehensive simulations. The performance-determining parameters, including the thickness of HTL and AL, as well as the doing and defect density ($N_A$ and $N_t$), were studied. Machine learning tools were employed to predict the performance metrics of solar cells. It was shown that $CsSnI_3$ was a promising HTL candidate that could be replaced with costly and low-conductive Spiro-OMeTAD. A power conversion efficiency of 23.9% was foreseen. It was shown that a thickness of 220 nm for the HTL with the doping and defect density of $5 \times 10^{19}$ cm$^{-3}$ and $5 \times 10^{17}$ cm$^{-3}$, respectively, yielded the highest PCE. Analysis of the effect of perovskite material, including $CsPbI_3$, $FAPbI_3$, and $MAPbI_3$, revealed that $FAPbI_3$ exhibited a higher efficiency than the others. Further analysis of the effect of thickness and doping density of AL determined that a PCE of 24.1% can be achieved with an AL thickness of 530 nm and $N_t = 10^{14}$ cm$^{-3}$. The ML approach provided a better understanding of the deciding factors while designing and producing the solar cell. The model achieved an accuracy score of 75% on the performance metrics of solar cells.

Although our survey comprehensively studied an advanced PSC structure with cost-effective and efficient HTL and AL, there are still some challenges. Interface defect engineering is a topic that requires a deep study. This problem can be addressed by introducing some new additives at the interface. The results of this study can be utilized to engineer perovskite composition for better performance. Introducing more inputs into ML can also be helpful in training a more precise and effective model.

**Author Contributions:** M.H.A. and S.A. (Samaneh Aynehband): Conceptualization, Visualization, Methodology, Validation, Formal analysis, Investigation, Data curation, Writing—Original Draft, Writing—Review and Editing. H.A. (Habib Abdollahi): Investigation, Writing—Original Draft, Writing—Review and Editing. H.A. (Homayoon Alimohammadi) and N.R.: Investigation, Formal Analysis, Data curation, Writing—Original Draft, Writing—Review and Editing. S.A. (Shayan Angizi): Investigation, Writing—Review and Editing. V.K. and R.T.: Writing—Review and Editing. R.M. and A.S.: Conceptualization, Writing—Review and Editing, Supervision, Project administration, funding acquisition. All authors have read and agreed to the published version of the manuscript.

**Funding:** A.S. acknowledge the foundation of INSF (GN: 95-S-48740 and 96016364) and Sharif University of Technology (GN: QA970816) and Niroo Research Institute (grant no. 98-50921-148).

**Institutional Review Board Statement:** Not applicable.

**Informed Consent Statement:** Not applicable.

**Data Availability Statement:** Not applicable.

**Acknowledgments:** The authors gratefully thank Marc Burgelman, University of Gent, Belgium, for providing the SCAPS-1D simulations software.

**Conflicts of Interest:** The authors declare no conflict of interest.

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
