# Peer review of "SCAPS Empowered Machine Learning Modelling of Perovskite Solar Cells: Predictive Design of Active Layer and Hole Transport Materials"

_photonics, doi:10.3390/photonics10030271_

Round 1

Reviewer 1 Report

This manuscript presents a simulation of perovskite solar cells with different HTLs and ALs with SCAPS-1D and machine learning. The authors found “CsSnI3 as the HTL possesses a negligible lattice-mismatch with perovskite, reducing the dangling bonds as recombination sites, high hole velocity, and low cost than commonly-used Spiro-OMeTAD.” However, serious mistakes may exist during the simulation process.

In Figure 3c, EQE spectrum shows that large portion of near infrared photons (wavelength ~ 900 nm) could be converted to electrons. As FAPbI3 has a bandgap of 1.51 eV, it should not have such capability. A basic rule of optoelectronics is semiconductors could not effectively absorb photons with energy less than its bandgap. Considering this, FAPbI3 should not have large absorption coefficient in the wavelength range > 850 nm. Thus, it also could not have large EQE in the same wavelength range. Lots of experiments confirms it. <ref. Adv. Mater. Volume28, Issue11, Pages 2253-2258; ACS Appl. Mater. Interfaces 2020, 12, 13, 15167–15174; Adv. Opt. Mater. Volume4, Issue11, Pages 1829-1837>. Besides, EQE and UV-vis absorption spectra in Figure 7 c, d also have similar mistakes. I strongly recommend that the authors check their simulation methods.

Reviewer 2 Report

In this manuscript, the authors employed SCAPS-1D and machine learning to study the characteristics of different HTLs (Spiro, CIS, and CsSnI3) and perovskite layers (FAPbI3, MAPbI3, and CsPbI3), including thickness, doping density, and defect density. CsSnI3 as the HTL possesses a negligible lattice-mismatch with perovskite, reducing the dangling bonds as recombination sites, high hole velocity, and low cost than commonly-used Spiro-OMeTAD. The machine learning  has become a powerful tool to discover new ways to approach the optimization of solar cells. I think this work would attract attention of the photonics’ readers. Therefore, I recommend the manuscript for publication in photonics after minor revisions. However, the following recommendations need to be considered:

1.      On Page 2, line 49-50, the authors claim: “In the last decade, a variety of materials in emerging technologies, such as optoelectronics, optical biosensors, and photocatalysis, have been introduced”. The development of the organic solar cell is very fast, the introduction section should be modified though citing recent references related studies to let readers know more clearly about the photovoltaic field. For example:

https://doi.org/10.1002/cjoc.202200652;  

https://doi.org/10.1016/j.optmat.2022.113219.

2. On Page 7, in Figure 3. c), the EQE spectrum of FAPbI3/CsSnI3 is obviously more than 900 nm, so the abscissa of this figure should be more than 1000 nm.

3. The author should carefully check and revise the whole manuscript, especially in the introduction section. I think many aspects of the description should be supported by recent references. And the latest achievements should be introduced. For example:

https://doi.org/10.1016/j.optmat.2022.113288;

https://doi.org/10.1016/j.optmat.2023. 113503.

Reviewer 3 Report

Please read and “fully” address the comments listed below:

1.              The ABSTRACT is not written in a logical order. Start with an overview of the topic and a rationale for your paper. Describe the methodology you used and the general outline of the manuscript. Also, in the end, state the result in more detail (i.e., provide some numbers).

2.              The novelty of your work is still unclear to the reader, which should be further detailed both in the Abstract and Introduction. In other words, the purpose of the research is missing, which must be clearly presented.

3.              It is mentioned that “Machine learning tools also affirm the extracted results with ~75% accuracy on the performance matrices of solar cells”, is this 75% ideal and conforms well to industry standards?

4.              Scale bars are missing from many figures. 

5.              Provide accuracy vs epoch graphs of train and test sets to determine the model’s accuracy and performance.

6.              Provide more explanation for this sentence: Page 10, Line 307: " At higher Nt values; however,  a sharp decline in VOC and JSC is seen.”.

7.              Similarly, this sentence needs to be better explained: Page 14, Line 386: “However, the figures of merit, including JSC, VOC, FF, and PCE, 386 decrease at a high concentration of defects”.

8.              Determine how the hyperparameters of your Random Forest model were optimized.

9.              The authors mentioned that the “prediction of the ML model determines a mean accuracy of 75% with a mean R2 of 0.78 between actual and predicted values per target feature”, and it was argued that “a fully trained model constructed on a wide range of input data (trained values) would result in more accurate and realistic outcomes. This statement is partially true because the dataset might be highly complex and even a large dataset may not significantly improve the model accuracy. In machine learning, this can refer to “Kolmogorov complexity” denoting the length of the shortest computer program that produces the output. Therefore, write a paragraph in your paper arguing that "reducing the complexity of your dataset" can potentially improve the accuracy of your model and reference the 2 papers listed below (they reduced the complexity of their dataset to dramatically improve the accuracy of their machine learning models)

·      Bolon-Canedo, V., & Remeseiro, B. (2020). Feature selection in image analysis: a survey. Artificial Intelligence Review, 53(4), 2905-2931.

·      Kabir, H., & Garg, N. (2023). Machine learning enabled orthogonal camera goniometry for accurate and robust contact angle measurements. Scientific Reports, 13(1), 1497.

10.     Conclusion: Can authors highlight future research directions and recommendations? Also, highlight the assumptions and limitations (e.g., shortcomings of the present study). Besides, recheck your manuscript and polish it for grammatical mistakes (you can use “Grammarly” or similar software to quickly edit your document).

Reviewer 4 Report

In this paper, authors have reported a perovskite solar cell and optimized the active layer and the hole transport layer to achieve high power conversion efficiency. It is reported that FaPbI3 is the best active layer with CsSnI3 as the hole transport layer. Further, the results obtained with the SCAPS-1d simulator are verified with the machine learning tools, showing a 75% match. The paper is capable of providing some direction in the analysis of perovskite solar cells and selecting suitable materials to achieve very high-power conversion efficiency. I have a few suggestions for the improvement of the manuscript.

1.       The EQE is quite high even above wavelength 820 nm for the CsSnI3 as the hole transport layer. Authors must provide some explanations. Is there any carrier generation taking place inside the CsSnI3 as well? In that case, how does the SRH recombination time affect the solar cell performance? Maybe authors should add the absorption spectrum of the hole transport layers as well.

2.        Line No. 274, it should be ‘doping density’. In 3(b) add the conduction band energy values for different layers as well as in Fig7(b) for the CsSnI3 layer.

3.       The explanation about the effect of doping density in the CsSnI3 layer on the efficiency parameters is not clear. How does the valance band affect the efficiency parameters and what is the importance of the quasi-fermi energy level difference between the hole and electron transport layer? The authors should provide a more clear and more elaborated discussion on that.

4.       How does the interface defect density affect the proposed solar cell? A simulation should be performed on that as well.

5.      The literature on the target applications of solar cell should be enlarged (see, i.e., Solar energy in space applications: review and technology perspectives. Advanced Energy Materials12(29), 2200125, 2022)

Round 2

Reviewer 1 Report

I recommend to accept this manuscript in the present form.

Reviewer 3 Report

The authors addressed my comments and the manuscript can be published in the current format.